# Students' Understanding of Microscopic Models of Electrical and Thermal Conductivity: Findings within the Development of a Multiple-Choice Concept Inventory

Lejla Jelovica [1,2], Nataša Erceg [3,*], Vanes Mešić [4] and Ivica Aviani [1,5]

1   Faculty of Science, University of Split, 21000 Split, Croatia; lejla.jelovica@uniri.hr (L.J.); iaviani@pmfst.hr (I.A.)
2   Faculty of Health Studies, University of Rijeka, 51000 Rijeka, Croatia
3   Faculty of Physics, University of Rijeka, 51000 Rijeka, Croatia
4   Faculty of Science, University of Sarajevo, 71000 Sarajevo, Bosnia and Herzegovina; vanes.mesic@gmail.com
5   Institute of Physics, 10000 Zagreb, Croatia
*   Correspondence: nerceg@phy.uniri.hr

**Abstract:** Solid-state physics has important applications in the development of modern technologies. Although solid-state concepts, such as electric and thermal conductivity, are included in the curricula at all educational levels, even university students have many incorrect ideas about them. The incorrect ideas identified in previous studies are mainly related to macroscopic aspects of solid-state physics. With the aim of gaining a more comprehensive insight into students' understanding of solid-state physics, we have developed a multiple-choice concept inventory on microscopic models of electric and thermal conductivity of solids (METCS). The inventory has been field-tested with a sample consisting of 233 first-year medical faculty and engineering students from the University of Rijeka (Croatia). METCS proved to have good psychometric features and it helped to uncover many incorrect ideas that have not been reported in the earlier physics education literature. The findings from this study could be a good starting point for the development of evidence-based, university-level tutorials on microscopic models of electric and thermal conductivity.

**Keywords:** physics education; conceptual understanding; concept inventory; electric conductivity; thermal conductivity; microscopic physical models; solids

## 1. Introduction

The development of many modern technologies would not be possible without solid-state physics [1]. Therefore, the understanding of phenomena such as thermodynamic and transport phenomena of solids is developing at all educational levels within the different physical models. This fact becomes particularly important currently, when, for example, the Center of Excellence for Semiconductors is being opened in Croatia [2]. This initiative is in line with the European Parliament's decision to improve the microelectronics sector as part of the further economic development of all EU countries [2].

Among the mentioned phenomena of conductivity, the concepts related to electrical and thermal conductivity are the most frequently covered in the curricula. Students start learning about them already in primary school. At the secondary school and university level, these concepts are covered again, but at a higher level of mathematization and abstractedness.

However, despite such good curricular coverage, educational research consistently identifies students' incorrect ideas [3–23], especially when it comes to microscopic models of conductivity [4–7,22,23]. When discussing students' conceptual understanding of conductivity, it is useful to introduce the following conceptual categories [3]: microscopic structure [3–11]; interparticle potential energy [3]; motion of particles at a certain temperature [2,3,5,14]; motion of free charge carriers in the external electric field [3,6,12,19];

electrical resistance/conductivity as a function of temperature [3,5,6,16] and thermal conductivity [3,4,17,18,20].

Many non-adequate ideas about electric and thermal conductivity simply result from the fact that the instruction failed to help the students develop functional mental models about conductivity and instead on mathematical derivations and solving of standard, quantitative problems [6,24]. Difficulties can also be the result of unclear textbook explanations of relevant concepts [5,6,25–29], and the generalisation of insufficiently clarified physical models [14,17,18,20,25,28,30]. Due to a lack of relevant conceptual models about conductivity, many students rely on phenomenological primitives when confronted with conceptual problems on conductivity, or simply try to apply mechanistic reasoning by transferring macroscopic world features to the microscopic context [31].

Teaching effectiveness may be improved through assessments, particularly through the use of concept inventories [32]. Concept inventories are research-based assessment tools that examine students' understanding of specific physics concepts [33,34]. By using a concept inventory in the classroom, it is possible to identify and overcome incorrect ideas, as well as to gain insight into students' conceptual structure and help them to develop valid mental models about physical phenomena [32,35].

It should be noted that there are already concept inventories [36,37] that partially assess students' understanding of the macroscopic phenomena of thermodynamics and the transport phenomena of solids. In particular, these existing tests in the areas of electricity and thermodynamics contain several tasks on the electrical and thermal conductivity of solids. However, these can be solved without knowledge of the microscopic physical models. For example, one task in the Thermal Concept Evaluation (TCE) inventory developed by Yeo and Zadnik [38], tests students' understanding of the concept of thermal conductivity of different materials from a macroscopic perspective. Specifically, students are asked to explain from experience why a metal ruler feels colder than a wooden ruler, without taking into account the microscopic view and the internal structure of these materials.

Moreover, the concepts covered in the mentioned conceptual tests do not correspond to the concepts presented in the Croatian curricula under the microscopic aspect. Therefore, our aim was to develop a new multiple-choice concept inventory and to use it for assessing university students' understanding of the Microscopic Models of Electric and Thermal Conductivity of Solids (METCS). It is based on the results of a previous study [3], and covers all concepts related to METCS included in the curricula of Croatia. By applying the test to a selected sample of Croatian students, we gained insight into their conceptual understanding of METCS.

## 2. Materials and Methods

We situated our research within the quantitative paradigm of educational research. In this section, we describe the development of the multiple-choice METCS concept inventory with a detailed overview of the concepts covered by the 1st version of the inventory, as well as its discrimination, item difficulty, validity and reliability. We also describe the sample of students who participated in the study and the curricula of the courses they took. Based on the above, the 2nd version of the test was developed (see Appendix A).

### 2.1. METCS Concept Inventory Design

Based on the literature review and the results of our initial research [3], we developed a multiple-choice concept inventory for measuring students' understanding of Microscopic Models of Electric and Thermal Conductivity of Solids (METCS). The inventory initially contained 31 multiple-choice questions grouped into 6 conceptual categories (Table 1).

**Table 1.** Conceptual groups, related concepts covered by the METCS concept inventory, ordinal numbers of tasks that examine the understanding of individual concepts, and presentation (marked with X) of the individual concepts in elementary school (ES), secondary school (SS) and university (U).

| Conceptual Groups | Concepts | Tasks | ES | SS | U |
|---|---|---|---|---|---|
| 1. Microscopic structure of solids | • Insulator structure | 1,2 | X | X | X |
| | • Conductor structure | 1,4 | X | X | X |
| | • Intrinsic semiconductor structure | 1,6 | X | X | X |
| | • Charge carriers in conductors and intrinsic semiconductors | 3,5,7,8 | | X | X |
| | • Doping and formation of extrinsic semiconductors | 9 | | X | X |
| | • Donor impurities | 10 | | X | X |
| | • Acceptor impurities | 11 | | X | X |
| | • Holes | 12 | | X | X |
| 2. Interparticle potential energy of solids | • Potential energy between two atoms of a solid body as a function of their distance | 13 | | X | X |
| | • Interparticle potential energy of the free charge carriers | 14 | | X | X |
| 3. Motion of particles in solids at a certain temperature | • Thermal motion of atoms in solids | 15 | X | X | X |
| | • Free-electron collisions in a conductor at room temperature | 16 | | X | X |
| | • Free-electron collisions in a conductor at low temperatures | 17 | | X | X |
| | • Motion of free electrons in a conductor without an external electric field | 18 | | X | X |
| | • Motion of free electrons in a semiconductor without an external electric field | 19 | | X | X |
| | • Motion of holes in a semiconductor without an external electric field | 20 | | X | X |
| | • Mobilities of electrons and holes in a semiconductor | 21 | | | X |
| 4. Motion of charge carriers in solids in the external electric field | • Movement of metal electrons in an external electric field | 22,23 | X | X | X |
| | • Movement of semiconductor electrons and holes in an external electric field | 24 | | X | X |
| | • Enhancement of the electrical conductivity by doping | 25 | | X | X |
| 5. Electrical conductivity/resistance of solids as a function of temperature | • Electrical resistance of metal as a function of temperature | 26 | | X | X |
| | • Electrical resistance of intrinsic semiconductor as a function of temperature | 27 | | | X |
| 6. Thermal conductivity of solids | • Thermal conductivity of conductors | 28 | X | X | X |
| | • Thermal conductivity of insulators | 29 | X | X | X |
| | • Heat transfer and change in the molecular kinetic energy of solid objects in direct contact | 30 | | X | X |
| | • Heat transfer and change in interparticle potential energy of solid objects in direct contact | 31 | | X | X |

Each question has four response options, only one of which is correct. The other three response options were distractors. The distractors have been designed based on the most common incorrect answers given by students who were administered the open-ended version of the METCS concept inventory [3].

The first twelve questions from the METCS inventory fall into the conceptual category of the microscopic structure of solids. These questions are designed to test whether the students understand the following: the structure of insulators (1,2), conductors (1,4) and semiconductors (1,6); the carriers of electric current in insulators (3), conductors (5) and semiconductors (7,8); the formation of extrinsic semiconductors (9); donor (10) and acceptor impurities (11); and the concept of holes (12).

In the second conceptual group, we examine the understanding of the potential energy of solid particles when they are constituent particles (13) and when they are free charge carriers (14).

The questions in the third conceptual group test students' understanding of the motion of solid particles at a certain temperature. The group includes questions on the following: the thermal motion of molecules of a solid (15); collisions of free electrons in a conductor at room temperature (16) and at low temperatures (17); the motion of charge carriers without the influence of an external electric field in a conductor (18) and in a semiconductor (19); the motion of holes in a semiconductor (20) and the mobility of electrons relative to holes in semiconductors (21).

The fourth group of tasks investigated the conceptual understanding of the following: the movement of free electrons in a conductor under the influence of an external electric field (22,23); the movement of free electrons and holes in a semiconductor under the influence of an external electric field (24); and the influence of doping on the increase in semiconductor conductivity (25).

The fifth conceptual group raises questions about electric resistance as a function of temperature in metals (26) and pure semiconductors (27).

The last four questions of the sixth conceptual group examine the students' understanding of: thermal conductivity of conductors (28) and insulators (29); heat transfer phenomena and the changes in the kinetic (30), or potential energy (31) of molecules of solids when they are in direct contact.

### 2.2. Participants

Convenience sampling has been used to obtain a sample of study participants. Concretely, our sample consisted of 233 first-year undergraduate students from the University of Rijeka. In total, 27 students (11.59%) were from the Medical Faculty and 206 (88.41%) were from the Faculty of Engineering. In the Faculty of Engineering subsample 45.49% were computer science students and 42.92% were electrical engineering students.

All students from our sample have learnt about the microscopic models of the electric and thermal conductivity of solids during their education. A total of 28% of the METCS concepts are already covered by primary school (through 8 years) curricula in Croatia, while even 88% of METCS concepts are covered by secondary school curricula [39,40]. When it comes to university curricula from the sampled faculties [41–43], it is important to note that they cover all the METCS concepts (Table 1). Concretely, the sampled students have learnt about the METCS concepts in at least one of the courses listed in Table 2 and were tested after completing the course.

During the first semester, students of the university integrated undergraduate and graduate study of Pharmacy (PHR) from the Medical Faculty, attended the Physics for Pharmacists course, which consisted of 30 h of lectures (L), 30 h of auditory exercises (AE) (Auditory exercises or traditional recitations are tied to the lectures and are led by a teaching assistant who teaches 30 to 50 students. The assistant chooses numerical problems to which the theory from the lectures is applied. Students present their own ideas for possible solutions to the assigned problems, which are modelled on the blackboard.) and 15 h of practical exercises (PE) (Table 2). Out of the total of 75 lessons, 6 lessons (6L) were devoted to the concepts of electric and thermal conductivity of solids within the context of the following topics: quantum mechanical description of atoms; chemical bonds; a macroscopic matter in a condensed state; thermal motion; internal energy; transport phenomena.

**Table 2.** Description of curricula and programmes related to the courses attended that include concepts on microscopic models of electrical and thermal conductivity of solids. For each group of participants, we provide information on the name of the group and the number of participants; the name of the university study program; the name of the course and the total number of hours devoted to lectures (L), auditory exercises (AE) and practical exercises (PE) for each course; the number of class hours devoted to the concepts of electrical and thermal conductivity of solids (ETCS).

| Name of the Group and Number of Participants | Name of the University Study Programme | Name of the Course, the Number of Hours (L, AE, PE) and Place in Curriculum | Number of the Class Hours Devoted to ETCS Concepts |
|---|---|---|---|
| PHR ($N_{PHR}$ = 27) | University integrated undergraduate and graduate study of Pharmacy | Physics for Pharmacists (30L + 30AE + 15PE) 1st year, 1st semester | 6L |
| CSC ($N_{CSC}$ = 106) | University undergraduate study of Computer Science | Introduction to Physics (30L + 30AE) 1st year, 1st semester | 6L |
| ELC ($N_{ELC}$ = 100) | University undergraduate study of Electrical Engineering | Physics II (30L + 30AE) 1st year, 2nd semester | 6L |

When it comes to students of the university undergraduate study of Computer Science (CSC) from the Faculty of Engineering, in the first semester they attended the Introduction to Physics course, which consisted of 30 hours of lectures and 30 hours of auditory exercises. This introductory physics course covered concepts related to the phenomena of electric and thermal conductivity of solids within the context of the topic area, the "structure of matter and motion of particles". A total of 6 lessons (6L) were dedicated to teaching about these concepts (Table 2).

Finally, students of the university undergraduate study of Electrical Engineering (ELC) from the Faculty of Engineering attended the second semester of the Physics II course, which consisted of 30 h of lectures and 30 h of auditory exercises. A total of 6 lessons (6L) covered the concepts of thermodynamic and transport phenomena of solids within the context of the topic area, "structure of matter and physical properties of matter—electric and thermal conductivity".

For all three groups of students from our sample, the lectures on concepts from Table 1 were given by the same teacher (co-author of this paper). For all groups, the part of the curriculum devoted to the METCS concepts was the same. The corresponding lectures lasted six hours in total. Concretely, the same textbook [44] contents were taught and the same traditional instruction was used.

*2.3. Procedures*

Testing of the 1st multiple-choice version of the MECTS inventory has been implemented after completing the above-mentioned courses in the academic year 2022/23. A paper-and-pencil method of test administration was used and the students were given 45 min to solve the test.

Afterwards, a SPSS database was created, where, for all 233 students from our sample, their answers to each of the 31 tasks were recorded. Also, a recoded version of that database has been created in which students' answers were coded as "1" if correct, and "0" in all other cases. These two databases were used for all statistical analyses in our study.

We analysed the properties of the METCS concept inventory using the computer programme for statistical data analysis IBM SPSS, version 25. This programme was used to check the normality of the distribution of the results by applying the Kolmogorov–Smirnov statistical test and to determine the item difficulty index and the reliability of the test. The

Microsoft Excel programme was only used to determine the discrimination index (ID) for each item according to Kelley's method [45].

Our descriptive study was conducted in accordance with ethical standards, including informed consent from all participants.

### 2.4. Item Discrimination Index and Item Difficulty

A discrimination index (ID) was determined for each item and it has been shown that tasks Q7, Q14, Q15 and Q31 have a discrimination index lower than 0.19 (Figure 1). In line with Cohen's instructions [35], we removed these items from the METCS. In this way, we have obtained the 2nd multiple-choice METCS concept inventory (see Appendix A). As a result of this statistical analysis, however, only one task from the second concept group of the first version of the METCS concept inventory remained (Task 13, Table 1). This task investigates the understanding of the concept of potential energy between two atoms of a solid body as a function of their distance. We have decided to place it in the conceptual group Motion of particles in solids at a certain temperature because the potential energy between two atoms of a solid is closely related to the movement of particles in the solid at a certain temperature. This is because the amplitude of the lattice vibrations in a solid increases with increasing temperature. The atoms move more strongly and are displaced from their equilibrium positions, which they had at a temperature of 0 K. The mean distance of adjacent atoms increases, as does the interparticle potential energy.

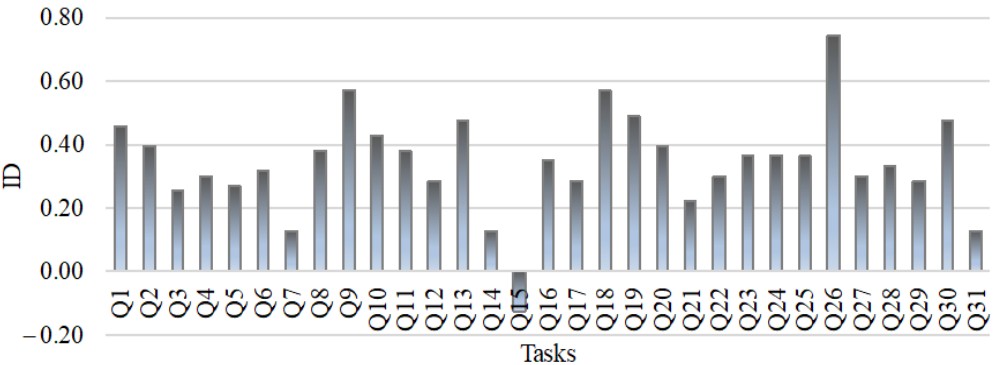

**Figure 1.** Item discrimination index of METCS concept inventory.

In this way, we obtained five conceptual groups in the second version of the METCS test. As already mentioned, these groups were formed with the aim of allowing a more transparent analysis and discussion of the results obtained, without compromising the statistical analysis in terms of validity and reliability.

Thereafter, we calculated the item difficulty index (proportion correct) for the remaining 27 multiple-choice questions. Table 3 shows that all the item difficulty indices lie in the recommended interval between 0.2 and 0.8 [46]. The average value of item difficulty amounts to 0.40 and is therefore slightly below the optimal average value of 0.50.

**Table 3.** Item difficulty index (*N* = 233).

| Item | Q1 | Q2 | Q3 | Q4 | Q5 | Q6 | Q7 | Q8 | Q9 | Q10 | Q11 | Q12 | Q13 | Q14 |
|------|------|------|------|------|------|------|------|------|------|------|------|------|------|------|
| mean | 0.64 | 0.42 | 0.66 | 0.36 | 0.32 | 0.33 | 0.61 | 0.39 | 0.34 | 0.34 | 0.31 | 0.54 | 0.53 | 0.51 |
| Item | Q15 | Q16 | Q17 | Q18 | Q19 | Q20 | Q21 | Q22 | Q23 | Q24 | Q25 | Q26 | Q27 | |
| mean | 0.50 | 0.31 | 0.21 | 0.24 | 0.24 | 0.36 | 0.39 | 0.41 | 0.48 | 0.24 | 0.30 | 0.28 | 0.44 | |

*2.5. Validity and Reliability Evidence*

Content validity evidence has been collected through expert interviews. Concretely, we interviewed 6 professors who had experience in teaching about microscopic models of electric and thermal conductivity, at the university level. In accordance with the guidelines of Adams and Wieman [34], the experts were administered the 31 METCS multiple-choice questions as well as Table 1, and they were asked to

1.  Check whether they can agree that the individual tasks can be associated with the given concepts and categories, as has been displayed in Table 1.
2.  Check whether the tasks have been clearly and comprehensibly formulated.
3.  Check the correct answer and examine whether the alternative answers are correctly defined.

They found that the inventory tasks cover all the concepts that are the subject of the research and that they are clearly and comprehensibly formulated, which makes them suitable for being used with the university population. All experts agree on the correct answer and also that the alternative answers are adequately defined. Thus, in agreement with Adams and Wieman [34], we concluded that content validity is confirmed.

Then, we checked the reliability of the METCS scores for all 31 tasks. As a measure of reliability, the Cronbach's alpha coefficient has been used. The preliminary statistical results showed the minimal reliability of the test ($\alpha$ = 0.693). A further analysis of the item-total statistics indicated the deletion of tasks Q7, Q14, Q15 and Q31, confirming the conclusion we had drawn from the determination of the item discrimination index regarding the deletion of the above-mentioned items. After this intervention, we obtained a higher Cronbach's alpha ($\alpha$ = 0.740), which, according to Cohen et al. [35], means a good reliability of the test. We therefore removed these four tasks from the METCS. Following Cohen's [35] instructions, the 1st version of the inventory was reduced to 27 questions (see Appendix A), and we performed further discussion with this 2nd version of the test. For this reason, our results and discussion consist of five conceptual groups related to solids: (i) microscopic structure; (ii) motion of particles at a certain temperature; (iii) motion of charge carriers in the external electric field; (iv) electrical conductivity/resistance as a function of temperature; and (v) thermal conductivity.

## 3. Results and Discussion

Our results were not normally distributed (see Table 4), so, in accordance with the recommendations of Cohen [35], we carried out a descriptive analysis of the response frequencies.

**Table 4.** Results of the Kolmogorov–Smirnov test for normality of the data distribution.

|  | **Statistic** | **df** | **Sig.** |
|---|---|---|---|
| total score | 0.138 | 233 | 0.000 |

We report and discuss the university students' responses to each of the 27 multiple-choice questions from the second version of the METCS concept inventory. These responses were grouped across the conceptual categories (see Figures 2–6).

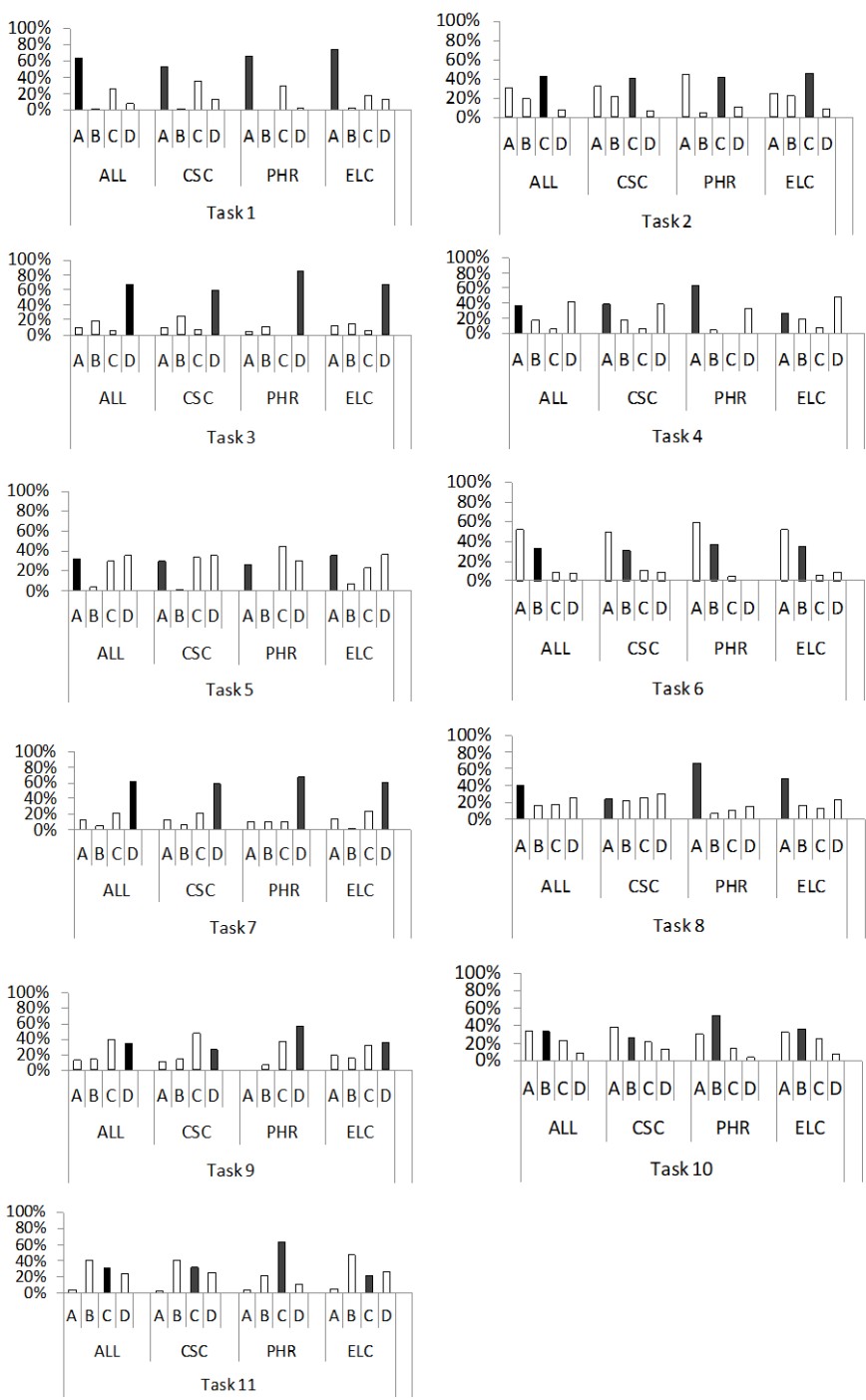

**Figure 2.** Percentage distribution of students' answers (A–D) to the questions from the conceptual category "microscopic structure of solids" (tasks 1–11), in the entire sample of respondents (ALL) and individual subsamples (CSC, PHR, ELC). Correct answers are represented by black bars.

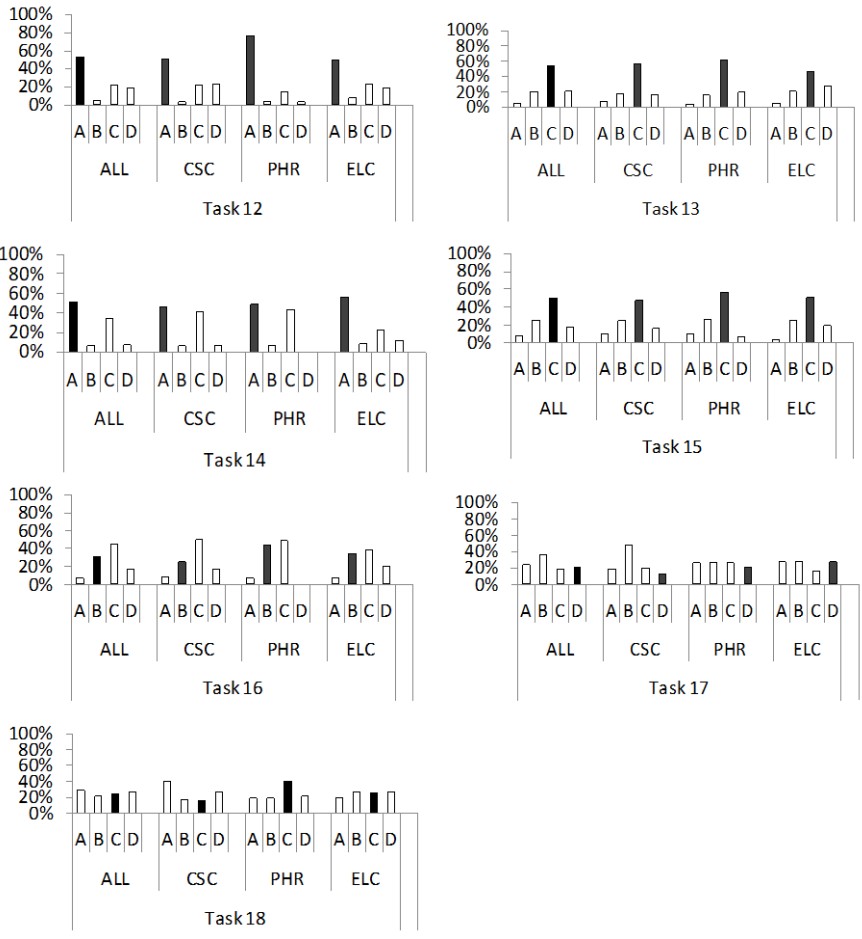

**Figure 3.** Percentage distribution of students' answers to the questions (tasks 12–18), from the conceptual category "motion of particles in solids at a certain temperature". For a detailed description, see Figure 2.

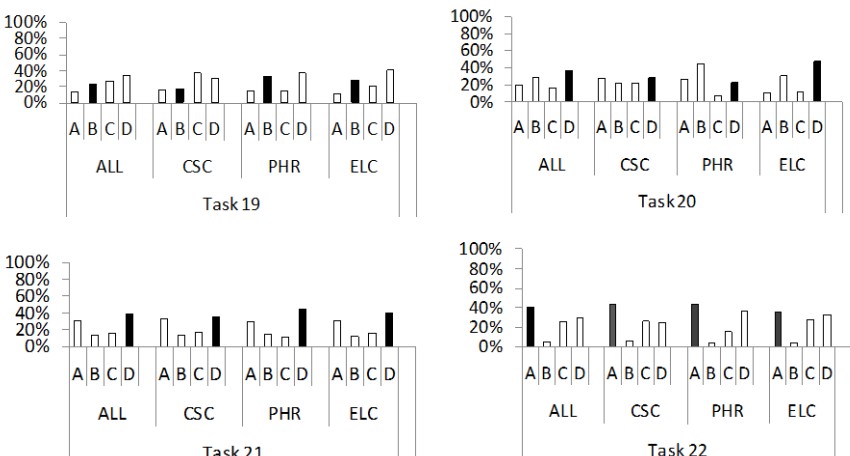

**Figure 4.** Percentage distribution of students' answers to the questions from the conceptual category "motion of free charge carriers in solids in the external electric field", (tasks 19–22). For a detailed description, see Figure 2.

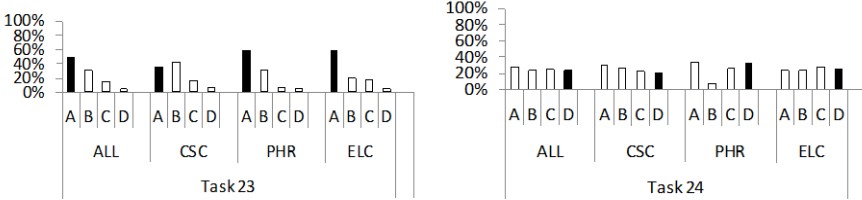

**Figure 5.** Percentage distribution of students' answers to the questions from the conceptual category "electrical resistance/conductivity of solids as a function of temperature" (tasks 23 and 24). For a detailed description, see Figure 2.

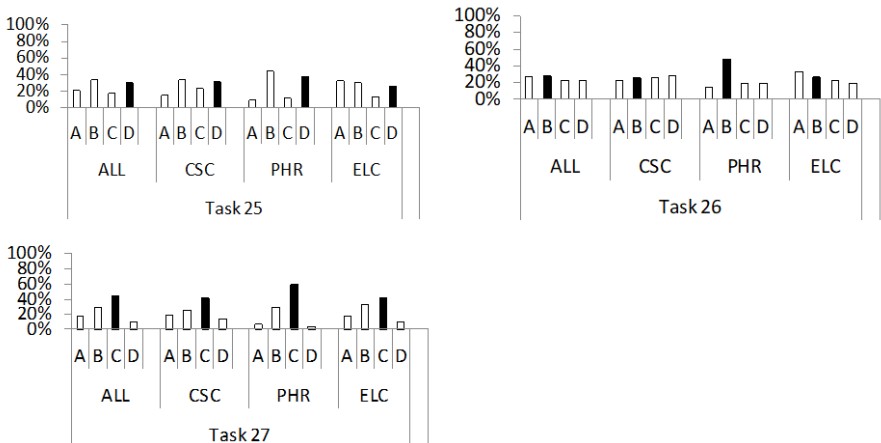

**Figure 6.** Percentage distribution of students' answers to the questions from the conceptual category "thermal conductivity of solids" (tasks 25–27). For a detailed description, see Figure 2.

### 3.1. Microscopic Structure of Solids

Concepts related to the microscopic structure of solids are covered by the first eleven tasks of the METCS concept inventory (second version). With these tasks, we tested students' understanding of the structure of insulators (1,2), conductors (1,4) and semiconductors (1,6); charge carriers in insulators (3), conductors (5) and semiconductors (7); the formation of extrinsic semiconductors (8); donor impurities (9); acceptor impurities (10) and the concept of holes (11). Figure 2 shows the percentage distribution of students' answers (A–D) to the multiple-choice questions 1 through 11.

When it comes to task 1, most students from all groups correctly answered that the structure of insulators, conductors and semiconductors can be crystalline or amorphous. However, it is interesting to note that in the distribution of answers, distractor C stands out as the most frequently chosen in all groups (ALL, 26%; CSC, 34%; PHR, 30%; ELC, 17%). This distractor reflects the incorrect idea that the structure of conductors and semiconductors is always crystalline, while the structure of insulators is amorphous. A possible explanation for the relatively large percentage of students choosing distractor C may be found in the transcripts of oral interviews that have been conducted in our initial research. For example, in our initial research [3], one student stated that: "Semiconductors have a crystalline structure because they have the same concentration of holes and electrons". The equal concentration of holes and electrons is apparently emphasised as a "regularity" in the structure of the semiconductor, which is falsely attributed to the crystal structure. This and other answers from the initial research (e.g., "I think insulators have an amorphous structure because they have an irregular structure, with irregularly bonded atoms"; "Conductors have a crystalline structure because their atoms are connected by regular bonds") indicate that crystalline, e.g., the amorphous structure of solids, is associated exclusively with regular or irregular structure, which is not surprising since a similar description of the structure of matter is used in Croatian textbooks [47]. At the same time, the concept of amorphousness is not additionally interpreted. Consequently, students ignore the fact that

amorphous substances are those that do not have a specific melting point but have a range of melting temperatures determined by their internal structure [44], which obviously leads to difficulties in the conceptual understanding of the structure of solids.

Tasks 2, 4 and 6 were used to assess students' conceptual understanding of the structure of crystal lattices of solids. In task 2, the most frequently chosen response option was the correct option, C, which states that the atoms of the insulator lattice are connected by covalent or ionic bonds. The fourth question about the structure of a metal lattice was answered correctly by the majority of students in the PHR group (63%), while in the other groups, the correct answer (A) was second in terms of frequency of selection (ALL, 36%; CSC, 38%; ELC, 27%). It claims that the crystal lattice of the conductor consists of positive ions connected by free electrons. Task 6 tested students' knowledge of the structure of a semiconductor lattice. The correct answer, B, which states that the crystal lattice of a semiconductor consists of atoms connected by covalent bonds, was the second most frequently chosen answer in all groups (ALL, 33%; CSC, 31%; PHR, 37%; ELC, 35%). The relatively high percentage of correct answers to questions 2, 4 and 6 could be related to the fact that the corresponding concepts, apart from the university textbook [44], are already represented in the Croatian textbooks for primary school [48–50], and for secondary vocational schools, when considering semiconductors [51,52].

On the other hand, the most common incorrect answer to the second question in all groups was distractor A (ALL, 31%; CSC, 32%; PHR, 44%; ELC, 24%). This states that the covalent bond is the only chemical bond between the atoms of the solid lattice of the insulator, which is a well-known incorrect idea in the literature [3,5]. The CSC and ELC groups had a lower percentage of incorrect answers compared to the PHR, which could be due to the fact that they had also learnt about the structure of insulators in other courses such as Materials Technology or Electronics I [53]. Some students from the initial research cite the covalent bonding of the atoms in the insulator as the cause of its poor conductivity. An example is the statement of one student interviewed: "In insulators, the atoms are tightly bound together by covalent bonds. . .therefore they do not move outside their frame, so the insulator does not conduct electricity" [3]. This indicates a misunderstanding of the electrical properties of insulators. This statement contradicts the fact that the majority of students correctly answered the third question about charge carriers in an insulator. They know that there are almost no charge carriers, i.e., electric currents, in the insulator, but they wrongly associate the electrical conductivity of the insulator with covalent bonds. This points directly to the fact that conceptual difficulties related to the conductivity of solids originate in understanding the structure of the substance itself.

The most common incorrect answer to question 4 in all groups (ALL, 41%, CSC, 38%; PHR, 33%; ELC, 47%) was distractor D. It was also the most common choice of the entire sample as well as the CSC and ELC groups. It contained the statement that the solid metal lattice consists of positive and negative ions connected by a metal bond. By choosing negative ions instead of electrons, the students show a misunderstanding that is consistent with the misconception discovered in the study [14], conducted by Ali et al. Concretely, Ali et al. [14] state that the identification of electrons with negative ions could be a consequence of the use of corresponding concepts in physics teaching, such as the "flow of electrons" and "flow of electrically charged atoms, i.e., ions", which are not adequately discussed in the interpretation of electric current in metals and electrolytes.

The most common answer to the sixth question in all groups was distractor A (ALL, 52%, CSC, 50%; PHR, 59%; ELC, 52%). It falsely claims that the solid lattice of semiconductors consists of electrons and holes, as research [7] shows. The reason for the high prevalence of this misconception in our sample could be didactic, due to ambiguities in the textbook texts. This is supported by the fact that, for example, in the textbook for secondary vocational schools [51], from which 53% of our respondents graduated, the electrical properties of semiconductor materials are described using the concepts of valence electrons and holes outside the context of the semiconductor structure.

In question 3, the majority of respondents in all groups chose the correct answer, D, which states that there are almost no free charge carriers in the insulator (ALL, 67%, CSC, 59%; PHR, 85%; ELC, 68%). The most frequently chosen distractor in all groups (ALL, 19%; CSC, 25%; PHR, 11%; ELC, 15%) was B. It reflects the erroneous idea that the charge carriers in an insulator are free electrons. This incorrect answer indicates that many students mix-up insulators with conductors [13]. Concretely, Santana et al. [13] state that some students believe that free electrons are present in both types of materials (conductors and insulators) because they mistakenly equate the concept of electrostatic induction in conductors with the concept of polarization in insulators (dielectrics).

Compared to question 3, students were much less successful in correctly answering question 5 (ALL 32%, CSC, 30%; PHR, 26%; ELC, 35%), which assessed whether the students understood that free electrons are the carriers of electric current in metals. In all groups, the most frequently chosen answers were distractors C (PHR, 44%) or D (ALL, 35%, CSC, 35%; ELC, 36%).

These distractors reflect the idea that free charge carriers in the conductor are generated by energetic excitation or by the influence of an external electric field. Our findings are in line with the results of earlier research reported by Wittmann et al. [19], who pointed out that some students believe that electrons can only move through the conductor if they are released from the atom with the help of energy, power or battery voltage. This is an indicator of reasoning based on Ohm's p-prime "force as a mover", which implies that the body can only move under the influence of an external force [31]. Such reasoning was also evident in the answer of an interviewed student from our initial research: "Electrons float in place, they do not collide and they do not transfer energy. . . they stand still because no force acts on them" [3].

Question 7 was "Why do free electrons in a pure semiconductor, as opposed to those in a metal, form pairs with holes?" The majority of respondents in all groups chose the correct answer, D (ALL, 61%, CSC, 59%; PHR, 67%; ELC, 60%), which contains the statement that free electrons already exist in metals, whereas in pure semiconductors they are only created by the excitation of the electron–hole pair. Among the incorrect answers, distractor C was chosen most frequently (ALL, 21%, CSC, 21%; PHR, 11%; ELC, 24%). It contains the statement that the covalent bond between the atoms of a semiconductor is weaker than in a conductor, so holes form in it. By choosing this distractor, the students show a lack of understanding of covalent bonding in accordance with the results of the study by Ünal et al. [54]. These authors noted that students incorrectly defined a covalent bond as a transfer of electrons between atoms, rather than a bonding of atoms through shared electron pairs [44]. Furthermore, the authors noted that students also incorrectly concluded that holes in semiconductors are created by the release of electrons from a weak covalent bond and their transition from atom to atom.

All these findings are in line with some ideas we identified through oral interviews, such as [3] "In semiconductors, a weak covalent bond is formed from which the electron can be released, leaving a hole. . . inside the conductor the bond is stronger. . . ", which also fits this misunderstanding.

In task 8, the most prevalent answer was the correct answer (A), (ALL (40%), PHR (67%), ELC (48%)), which states that semiconductor doping is the replacement of crystal lattice atoms by atoms with a similar electronic structure. On the other hand, answer D stands out as the most frequently chosen distractor in all groups (ALL, 25%; CSC, 30%; PHR, 15%; ELC, 23%). It reflects the idea that doping is the process of adding electrons to fill holes as a kind of defect in the crystal lattice. This wrong idea about doping is followed by the wrong idea about holes, which is consistent with earlier research findings [5]. Concretely, Garcia and Criado [5] point out that their respondents consider holes to be evidence of damage to the crystal structure of the semiconductor material. Another possible reason for choosing the distractor D instead of the correct answer (A) may be terminological [24,55], due to the lack of distinction between the terms "substitute" and "addition". This argument is supported by findings from our interviews, where some

students claimed that [3] "...doping is a substitution, i.e., the addition of some atoms to the crystal lattice of a semiconductor...".

Tasks 9 and 10 tested students' understanding of the concepts of donor and acceptor impurities used for doping semiconductors.

The distribution of answers to question 9 shows that the majority of respondents in the PHR (56%) and ELC (35%) groups correctly concluded that donor impurities are neutral atoms with more valence electrons compared to semiconductor atoms (answer D). At the same time, this answer was the second most common answer in the groups ALL (34%) and CSC (27%). Furthermore, the most frequently chosen distractor in all groups (ALL, 39%; CSC, 48%; PHR, 37%; ELC, 31%) was answer C, which contains the statement that donor impurities are negative ions with an excess of electrons. This misconception is consistent with research findings [3,5,6]. For example, students cite "an excess of electrons relative to the holes" as the reason for the negative charge of donor impurities [3]. This shows that there are also misconceptions about holes, which we investigated in the 11th task.

When it comes to task 10, the correct answer was (B), which states that acceptor impurities are neutral atoms with fewer valence electrons than semiconductor atoms (ALL, 34%; PHR, 51%; ELC, 36%). Again, the most frequently chosen distractor was the same for all groups (ALL, 34%; CSC, 38%; PHR, 30%; ELC, 32%), which was the distractor A. It states that acceptor impurities are positive ions with electron deficiencies. The choice of this distractor suggests that for students it is difficult to understand the concept of a localised hole as an empty space in a covalent bond within an acceptor impurity atom formed in the doping process [56].

In task 11, the correct answer (C), which reflects the idea that holes are empty states in the electronic band that are created by electron excitation or doping, has been most frequently chosen only in the PHR group (63%). In the other groups, distractor B predominates (ALL, 41%, CSC, 40%; ELC, 47%). According to this incorrect answer, holes are empty spaces in the atoms of the crystal lattice that are created by the release of electrons. This wrong idea may be due to the fact that some authors of secondary school textbooks take an inappropriate approach to explaining the concept of a hole. For example, in the textbook [51], the generation of electron–hole pairs is described as a process that occurs by the breaking of covalent bonds during the ionisation of atoms of the crystal lattice of a semiconductor, rather than by the excitation of electrons from the valence band to the conduction band without "breaking of covalent bonds" [57]. In some high school textbooks [58], holes are also defined as "empty spaces in the valence band created by the release of valence electrons from a covalent bond". Additionally, some researchers interpret holes in a way that is not in line with some of the relevant textbooks [57]. Fayyaz et al. [6], for example, assume that holes formed by doping with acceptor impurities are neutral, which contradicts the definition of holes as positively charged particles in relevant textbooks such as the book by Kittel [57]. Considering the different interpretations of the term "hole" in the physics literature, it is not surprising that students experience difficulties in understanding what a "hole" is.

### 3.2. Motion of Particles in Solids at a Certain Temperature

Our second conceptual category includes seven tasks (12–18) that test students' understanding of the motion of particles in a solid at a certain temperature. They are related to understanding the potential energy of the constituent particles of a solid (12); collisions of free electrons in a conductor at room temperature (13) and at low temperatures (14); the motion of charge carriers without the influence of an external electric field in the conductor (15) and in the semiconductor (16); the motion of holes in the semiconductor (17) and the mobility of free electrons in the semiconductor relative to holes (18). The percentage distribution of the individual answers (A–D) to the multiple-choice questions in each of the four groups of respondents (ALL, CSC, PHR, ELC) is shown in Figure 3.

In task 12, the majority of students chose the graph that correctly depicts molecules at an equilibrium distance having a minimum potential energy that increases as the particles

move closer or further apart (ALL, 54%; CSC, 51%; PHR, 77%; ELC, 50%). This result is not surprising, as the concept of the dependence of interparticle potential energy on the distance between particles of solids is taught in the Croatian Physics [59] and Chemistry [60] subjects at the high school level, as well as in university courses such as Introduction to Physics [42] or Physics II [41]. The most frequent incorrect answers were C in the groups (ALL, 22%; PHR, 15%; ELC, 23%) and D (CSC, 22%). The distractor C reflects an incorrect idea that has already been reported in earlier studies [61,62]. Concretely, distractor C contains a graph showing a decrease in potential energy with increasing distance between solid particles. The fact that students often choose this distractor can be interpreted as a consequence of the activation of the phenomenological primitive "further is weaker" [31]. On the other hand, the selection of distractor D, where the curve of the correct shape is misplaced in the coordinate system, indicates that the students were trying to remember "something that was taught in class" [3]. However, this only indicates that they did not "store it with understanding" in the first place.

The distribution of answers for question 13 shows that most students in all groups (ALL, 54%; CSC, 57%; PHR, 62%; ELC, 47%) correctly concluded that free electrons collide with ions in the crystal lattice that are out of equilibrium. The most frequent incorrect answers were B (CSC, 18%) or D (ALL, 21%; PHR, 19%; ELC, 27%). These include claims that free electrons mainly collide with each other like gas molecules (B) or with the ions of the crystal lattice when the ions are in an equilibrium position (D). These wrong ideas may stem from the application of Drude's classical mechanical model of the conductivity of metals, which is probably not sufficiently discussed in class. The Drude model ignores the mutual interactions of the free electrons, as well as their interactions with the lattice ions [63], so the behaviour of the free electrons could be understood in this sense as analogous to the behaviour of the particles of an ideal gas. However, in contrast to the particles of an ideal gas, which collide with each other and thus achieve thermal equilibrium, the free electrons in the Drude model do not collide with each other, but with the ions of the crystal lattice. Another possible source of difficulties in understanding the motion of metal particles, which manifests itself in the selection of the distractor D, is the unrealistic hypothesis of the Drude model, which assumes that free electrons collide with ions of the crystal lattice that are static in their equilibrium positions [63]. In this case, the mean free path of the electron would be about ten times smaller than it is in reality. In fact, electrons move through molecular orbitals, electron bands that are created by connecting the atomic orbitals of atoms with valence electrons and extend through the entire crystal. Since electron bands are formed from atomic orbitals, the electrons in them are still integral parts of these atoms and do not collide with their positive ions in equilibrium positions.

The most common response to question 14 was the correct answer, A, which was chosen by about half of the respondents in each group (ALL, 51%; CSC, 47%; PHR, 49%; ELC, 53%). They knew that at low temperatures the collision of free electrons with lattice ions is significantly reduced due to the lower amplitude of ion oscillations. In contrast, when choosing distractor C, which was the second choice in all groups (ALL, 34%; CSC, 42%; PHR, 44%; ELC, 22%), respondents incorrectly concluded that free electrons collide less with lattice ions at low temperatures due to their lower kinetic energy. This incorrect idea is based on Drude's electron gas model, according to which the entire kinetic energy of free electrons comes from their thermal motion. In fact, the kinetic energy of the free electrons comes from the quantum mechanical motion in the atomic orbitals of an atom, i.e., from the motion in the electron bands created by chemical bonds in a crystal. These velocities are considerably higher than the thermal velocities, which becomes particularly evident at low temperatures. The mentioned incorrect idea could be a consequence of the students' misunderstanding of the structure of solids, as identified in the fifth task, according to which the existence of free charge carriers in the conductor is caused by energetic excitation. In this sense, low temperatures could falsely indicate a small number of free electrons with low kinetic energy and, consequently, a small number of collisions of free electrons with ions of the solid metal lattice. The immediate conclusion about a small

number of collisions based on a low temperature, i.e., low kinetic energy, may also be a consequence of the activation of the p-prime "motion dies away" [31].

In task 15, students from all groups most frequently chose the correct answer, C (ALL, 50%; CSC, 48%; PHR, 56%; ELC, 51%), which states that free electrons move through a metal crystal at high velocities of the order of $10^6$ ms$^{-1}$. On the other hand, distractor B, which contains the statement that free electrons in a metal move along atomic orbitals but at a greater distance from the atomic nucleus, was chosen by approximately 25% of students in each group (ALL, 25%; CSC, 25%; PHR, 26%; ELC, 25%). It seems that here the students ignored the fact that free delocalised electrons in the form of "electron gas" are common to the whole crystal [44], which could be a consequence of difficulties in their understanding of the conductor structure (see discussion of task 4).

Tasks 16 and 17 tested students' understanding of the concept of the movement of free electrons and holes in a semiconductor. The correct answer, B, to question 16, which included the statement that electrons in the conduction band of a semiconductor move freely like conduction electrons in a metal (with high velocities in all crystallographic directions), was the second most common choice in each group (ALL, 31%; CSC, 25%; PHR, 44%; ELC, 34%). Most students in all groups (ALL, 45%; CSC, 50%; PHR, 49%; ELC, 38%) selected the incorrect answer C to question 16, which included the statement that electrons in the conduction band of a semiconductor move with the constraints due to the higher occupancy of electronic states than in a metal. The truth is exactly the opposite: the state occupation limits the electronic conductivity in a metal, not in a semiconductor. However, according to the model of the electronic band structure, in the metal there are already free electrons in the range of the highest permissible energies, the so-called conduction band, while this band is completely empty in semiconductors. Therefore, the conduction electrons in a semiconductor are less constrained in their mobility than the conduction electrons in a conductor [64]. Relevant to this discussion may be also the fact that some students seem to mix up conduction and valence bands, as indicated by this quote from our initial research [3]: "Free electrons in the conduction band of semiconductors move only to a limited extent, because some states are occupied by holes". It should be noted that holes exist only in the valence band.

The distribution of answers to question 17 shows that students experience substantial difficulties in developing an understanding of hole movement. Concretely, only 21% of all students, most of whom belong to the ELC subgroup (28%), correctly answered that holes in a semiconductor move at a high speed similar to free electrons (D). The majority of students in the total sample (ALL, 36%) and in the CSC subgroup (48%) chose distractor B. It reflects the erroneous idea that holes in a semiconductor do not move without the influence of an external electric field. Such an idea possibly resulted from an activation of the p-prime "force as a mover" [31]. When it comes to distractor A, it was chosen most frequently in the PHR (26%) and ELC (28%) groups, with a slight percentage difference compared to the other answers offered. This indicates an arbitrary selection, i.e., that the corresponding wrong idea was not emphasised.

The correct answer, C, to question 18 was the most common answer in the PHR group (41%). It contains the statement that the mobility of the free electrons of a semiconductor is higher than that of the holes, which is due to the higher energy of the electrons in the conduction band compared to the electrons in the valence band. In other groups, the wrong answers of A (ALL, 29%; CSC, 41%) or D (ELC, 27%) were most frequently chosen. These distractors reflect the erroneous ideas that the mobility of the free electrons of semiconductors compared to the holes is (i) higher because the holes do not move without the influence of an external electric field (A) or (ii) higher for N-type semiconductors and lower for P-type semiconductors (D). The selection of answer A could be related to the incorrect idea that free charge carriers only move due to external stimulation, which we analysed within the discussion for task 5. On the other hand, the students' conclusion that free electrons are more mobile in N-type semiconductors than in P-type semiconductors, reflects the idea that a higher concentration of electrons is the cause of their "stronger"

mobility compared to holes. This idea is a possible consequence of the p-prime activation "bigger is stronger" [31].

### 3.3. Motion of Charge Carriers in Solids in the External Electric Field

Next, we will discuss the motion of free charge carriers in solids, under the influence of an external electric field. This conceptual category includes tasks 19 through 22, which contain questions on the following concepts: motion of free electrons in a conductor (19); instantaneous ignition of a light bulb by closing a circuit (20); movement of free electrons and holes in a semiconductor (21); and influence of doping on the increase in semiconductor conductivity (22). The percentage distribution of the individual answers (A–D) to questions from this conceptual group is shown in Figure 4.

In task 19, the correct answer was that the electrons in the conductor move in all directions, with the predominant direction being opposite to the direction of the electric field (answer B). However, only 24% of all students chose the correct answer. The students most frequently chose distractors C (CSC, 37%) or D (ALL, 35%; PHR, 37%; ELC, 40%). These incorrect answers reflect the ideas that for a metal under the influence of an external electric field, free electrons move in a straight line in the same (C) or opposite (D) direction to the direction of the electric field. These wrong ideas have been also reported by Nelson et al. [15]. Concretely, in that study, the respondents stated that the electrons move in a straight line by considering only the effect of the electric field force [15], which is a possible consequence of the activation of the p-prime, as "things move in the direction they are facing" [31].

Task 20 tested how students explain why a light bulb switches on almost instantaneously after the circuit is closed. The correct answer, D, is that a bulb lights up almost instantaneously because closing the circuit results in an electric field being established at the speed of light and that electric field causes the collective movement of free electrons. Answer D was the predominant choice in the groups (ALL 36%; CSC, 29%; ELC, 47%), while it was recorded as the third most frequent choice in the study group PHR (23%). The most frequently incorrect answers were A (CSC, 27%) or B (ALL, 28%; PHR, 44%; ELC, 31%). These distractors contain the claims that the light bulb lights up almost instantaneously because the free electrons are set in motion: (i) collectively by an electrical signal travelling at the Fermi speed (A) and (ii) at a drift velocity that directs the motion of the free electrons and is comparable to the speed of light (B). Students who chose answer A incorrectly associated the Fermi velocity with an electrical signal rather than with free electrons in a metal at absolute zero temperature. On the other hand, distractor B reflects the erroneous idea that drift velocity is the same kind of force (conceptual confusion) that is comparable to the speed of light (factual misconception).

The distribution of answers to question 21 shows that the concept of the movement of free charge carriers in a semiconductor under the influence of an external electric field was correctly interpreted by the majority of respondents in all groups (ALL, 39%; CSC, 36%; PHR, 44%; ELC, 40%). They chose the correct answer, D, according to which the charge carriers move in all directions, with the predominant direction being determined by the external electric field. The second most common answer in all groups (ALL, 31%; CSC, 33%; PHR, 30%; ELC, 31%) was distractor A, which states that electrons move against the direction of the electric field, while holes are stationary. This idea is consistent with the so-called hopping model [56], according to which electrons move from hole to hole. In this way, they fill empty spaces and at the same time free up new spaces in the lattice. The incorrect conception of "holes at rest in an external electric field" contradicts what we have observed in students' answers to question 17, where many students stated that the holes inside a semiconductor move under the influence of an external electric field. This result is consistent with Redish's context [65], as students choose different incorrect statements about the same concept depending on the task context, without having a conceptual understanding.

In task 22, the highest percentage of students in all groups (ALL, 41%; CSC, 44%; PHR, 44%; ELC, 36%) chose the correct answer A, which stated that the addition of an acceptor increases the conductivity of a semiconductor because the number of holes increases. At the same time, the most frequently chosen distractors were C (CSC, 26%) or D (ALL, 29%; PHR, 37%; ELC, 33%). These distractors reflect the ideas that doping a semiconductor with acceptor impurities increases its electrical conductivity due to an increase in the mobility of electrons in the conduction band (C) or due to a reduction in the energy gap between the valence and conduction bands (D). An interesting idea of a correlation between the conductivity of P-semiconductors and the mobility of electrons in the conduction band was described by an interviewed student [3], who assumed that a larger number of holes leaves more space for the movement of electrons. He used the analogy of a car that moves from point A to point B faster the fewer obstacles (which it would otherwise have to drive around) there are on the way between these points. Thereby, he compared the car to an electron and the path without obstacles to a valence band with a large number of holes. Furthermore, the emphasis on the "smaller energy gap" as the cause of the increased conductivity of P-semiconductors, could be a consequence of attempting to analyse the associated band diagram exclusively from a visual perspective, without conceptual understanding. A narrow acceptor band located directly above the valence band can give the visual impression of a reduction in the width of the forbidden band. However, the width of the forbidden band remains unchanged and the acceptor band receives electrons from the valence band, increasing the concentration of holes in the valence band and thus the conductivity of the semiconductor.

### 3.4. Electrical Conductivity/Resistance of Solids as a Function of Temperature

With this conceptual group, we examined students understanding of the temperature dependence of electrical resistance in metals (23), i.e., in pure semiconductors (24). The percentage distribution of students' answers (A–D) to these two questions is shown in Figure 5.

In task 23, the correct answer A was the most frequently chosen option in groups (ALL, 49%; PHR, 59%; ELC, 59%), while in group CSC (35%), it was the second most frequently chosen. It consists of the claim that an increase in temperature increases the electrical resistance of metals due to stronger vibrations of the ions, which disturb the movement of the free electrons. Under these circumstances, the average free path length of the electrons is reduced, which leads to a lower conductivity of the metal [56]. The most common incorrect answer was distractor B (ALL, 31%; PHR, 30%; ELC, 20%), which was also the first choice of students from the CSC group (42%). It reflects the erroneous idea that heating a metal decreases its resistance, i.e., its conductivity increases, because free electrons move faster. This idea is consistent with some previously discovered wrong ideas [3,6,16,19], where the respondents believed that heating a metal increases the kinetic energy of the free electrons and thus their velocity, resulting in a higher electrical conductivity of the metal. It seems that students from the abovementioned studies ignored the thermal excitation of the ions of the crystal lattice, which in reality has a dominant influence on the electrical conductivity of the conductor. The selection of distractor B is also a possible consequence of the activation of Ohm's p-prime: "increased effort–decreased resistance" [31]. According to this p-prime, the greater activity of the body, which can be described by the faster movement of the body, leads to a lower resistance of the medium through which the body moves. In this case, the free electrons represent the "bodies" and the metal lattice is the "medium".

With task 24, we tested students' conceptual understanding of the temperature dependence of electrical resistance in pure semiconductors. The correct answer (D) is that the decrease in the resistance of a pure semiconductor results from the increase in the number of free charge carriers (electrons and holes), (ALL, 24%; CSC, 20%; PHR, 33%; ELC, 25%). Distractors A (ALL, 28%; CSC, 31%; PHR, 34%) or C (ELC, 28%) were most frequently chosen. They contain the statements that the electrical resistance of a pure semiconductor: (i) decreases with increasing temperature due to the faster movement of charge carriers (A)

and (ii) changes inversely compared to metals due to a higher concentration of holes (C). The selection of distractor A indicates the existence of an incorrect idea similar to the one identified and discussed in the previous task, i.e., task 23. As with metals, students mistakenly believed that the conductivity or resistance of a semiconductor depends mainly on the kinetic energy (or velocity) of the charge carriers, which changes with temperature. The selection of distractor C, in which the increase in hole concentration in the semiconductor with increasing temperature is mentioned, indicates an understanding of the process of hole formation due to thermal excitation. However, in the same distractor, it is explained that the change in the concentration of holes (without the inclusion of electrons as free charge carriers) is the reason why the resistance of semiconductors changes inversely compared to conductors. According to some interviewed students [3], pure semiconductors only have holes as charge carriers, because they conduct electric current worse than doped semiconductors, which have both holes and electrons as free charge carriers. This idea is a possible consequence of the activation of the p-prime "more is stronger" [31], according to which more types of charge carriers result in a "stronger" electrical conductivity of matter.

### 3.5. Thermal Conductivity of Solids

The last three tasks of the METCS concept inventory test students' understanding of the thermal conductivity of conductors (25) and insulators (26); heat transfer between solids in direct contact at different temperatures (27). The percentage distribution of students' answers (A–D) to questions from this group is shown in Figure 6.

In task 25, the correct answer (D), which contains the claim that heat transfer in metal occurs through the transfer of kinetic energy from free electrons to ions, was chosen second most frequently in the ALL (30%), CSC (31%) and PHR (37%) groups, and third most frequently in the ELC (26%) group. At the same time, the most frequently chosen answers were distractors A (ELC, 32%) or B (ALL, 33%; CSC, 33%; PHR, 44%). These include the claims that heat transfer in a metal is caused by the transfer of (i) the vibrational energy of the ions of the solid-state lattice (A) and (ii) the kinetic energies between free electrons (B). The selection of distractor A could result from a misunderstanding of the structure of the metal. Concretely, the findings for task 4 show that a large number of our respondents mistakenly believed that a metal contains negative ions instead of free electrons in addition to positive ions. Such an idea can lead to a wrong idea in which ions are wrongly assigned the role of heat carriers in the metal, as they would be the only constituent particles of the metal. On the other hand, the selection of distractor B is a possible consequence of the wrong idea identified and discussed in task 13, according to which the students claimed that free electrons in a metal mainly collide with each other like gas molecules. This could have led them to the conclusion that the collisions of free electrons transfer kinetic energy between them, which in a macroscopic sense leads to the transfer of heat through the metal.

In task 26, the correct answer (B) is that insulators are worse conductors of heat than metals because the heat in insulators is mainly conducted by the vibrations of the lattice atoms, but not by free electrons (as there are almost none in insulators) (ALL, 28%; CSC, 25%; PHR, 48%; ELC, 26%). On the other hand, distractor A was the most frequently chosen answer in the ELC group (33%), while distractor (D) was the most frequently chosen in the CSC group (28%). These distractors contain claims that insulators conduct heat less well than metals because the heat in the insulator is conducted mainly due to (i) the vibrations of the lattice atoms, which are weaker due to the stronger interatomic bonds (A) and (ii) the free charge carriers, whose number is small (D). The selection of distractor A suggests that the students incorrectly compared the thermal conductivity of the insulator to the conductivity of the metal based on the "strength" of the vibrations of the lattice atoms, which may be a consequence of p-prime activation: "lower effort–higher resistance" [31]. This p-prime implies that the weaker activity of the particles within a substance, which can be observed in the context of weaker atomic vibrations, leads to a higher resistance or a lower conductivity of the substance. Distractor A is also an indicator of difficulties in understanding the structure of solids. Concretely, it claims that insulators have stronger

interatomic bonds than metals, although this is not always the case. By choosing distractor D, students have incorrectly assigned a dominant role to free charge carriers in determining the thermal conductivity properties of insulators. This is in contrast to the correct opinion of the majority of students expressed in the third task, that there are almost no charge carriers in the insulator. The contradictory ideas presented about the same concept in different contexts correspond to Redish's principle of context [65]. The selection of distractor D could also be a possible consequence of the p-prime activation "less is weaker" [31], according to which a small number of free charge carriers is the reason for the poor thermal conductivity of the insulator.

Finally, task 27 was about understanding the microscopic concept of heat transfer between solids at different temperatures that are in contact with each other in a thermally isolated system. The majority of students from all groups (ALL, 44%; CSC, 42%; PHR, 59%; ELC, 41%) chose the correct answer (C), which states that only the kinetic energy of the molecules is transferred to the contact surface by a collision of the body molecules. Approximately one-third of the respondents (ALL, 29%; CSC, 26%; PHR, 30%; ELC, 32%) chose distractor B. This answer reflects the incorrect idea that the collision of the two bodies' molecules results in a transfer of internal energy through the contact surface of the two bodies. This incorrect idea is consistent with findings from earlier research [3,61,66], according to which the concept of internal energy is often equated with the concept of kinetic energy. At the same time, the intermolecular potential energy of a solid body, which is modelled according to the ideal gas model [3,61], is unjustifiably neglected.

## 4. Conclusions

In everyday life, we are surrounded by many modern technologies that were developed based on knowledge of solid-state physics, which is why an understanding of the most important aspects of solid-state physics can be considered an important part of scientific literacy. Consequently, students learn about selected solid-state concepts, such as electric and thermal conductivity, throughout all levels of education. In order to improve the quality of learning and teaching about solid-state physics, it is important to identify the actual learning outcomes for the current curricula. Earlier research uncovered many incorrect ideas about macroscopic aspects of electric and thermal conductivity, and, in this study, we aimed to develop a better understanding of how university students reason about microscopic aspects of conductivity.

To that end, a microscopic models of electric and thermal conductivity of solids (METCS) concept inventory has been developed. It has been shown that METCS can be reliably used to measure students' understanding of (i) microscopic structure, (ii) motion of a particle at a certain temperature, (iii) motion of free charge carriers under the influence of an external electric field, (iv) electric resistance or conductivity as a function of temperature, and (v) thermal conductivity.

This research not only further corroborated findings about students' incorrect ideas from earlier research, but also uncovered some new incorrect ideas. When it comes to corroboration of earlier research, it is mainly related to the structure of solids. Concretely, students mistakenly believe that the covalent bond is the only possible bond between insulator atoms [5] or that it is weaker in semiconductors than in conductors, so that semiconductor holes form in them [54]. They also mistakenly believe that the lattice of metals consists of positive and negative ions [14] and that the lattice of semiconductors consists of electrons and holes [5]. In addition, many students believe that free charge carriers (electrons) also exist in insulators [13] and that they are generated in conductors by energetic excitation [19]. Regarding the structure of extrinsic semiconductors, students mistakenly think that impurities are ions [5,6], and they incorrectly associate a decrease in the interparticle potential energy with an increase in the distance between the particles of a solid [61,62]. Many students erroneously think that applying an electric field across metals results in linear motion of the free electrons [6,15], and they associate the temperature dependence of the electric conductivity of metals and semiconductors with the velocity of

motion of the charge carriers [6,16,19]. When heat is transferred between bodies at different temperatures, students have the incorrect idea that the internal energy is transferred at the contact surface [61,66].

The newly discovered incorrect ideas, presented in this paper, belong to different conceptual categories. When it comes to their ideas about the structure of solids, many students mistakenly believe that conductors and semiconductors have an exclusively crystalline structure, while insulators have an amorphous structure. They also often believe that holes are empty spaces in the atoms of the crystal lattice created by the release of electrons and that doping is the process of adding electrons to fill the holes. Some students also confuse free electrons in a metal with particles moving on atomic orbitals that usually collide with each other like gas molecules or with ions of a crystal lattice when the ions are in equilibrium. The students believe that the lower kinetic energy of free electrons is the main reason why there are fewer collisions between free electrons and ions at low temperatures. They also often have the incorrect idea that conduction electrons in a conductor can move more "freely" than conduction electrons in a semiconductor due to the lower occupancy of the electronic states in the conduction band of conductors compared to the conduction band of semiconductors. On the other hand, many students erroneously think that the mobility of free electrons in semiconductors is generally higher than that of holes that do not move, or that the mobility of free electrons relative to holes is higher in N-type semiconductors than in P-type semiconductors. When it comes to the motion of free charge carriers in solids under the influence of an external electric field, students often believe that the current flow in the light bulb occurs instantaneously because the free electrons move collectively due to an electric signal that propagates at a Fermi or drift velocity comparable to the speed of light. Moreover, many students mistakenly think that doping semiconductors with acceptor impurities increases the electric conductivity of the semiconductor due to an increase in the mobility of electrons in the conduction band or a decrease in the width of the forbidden band. On the other hand, they explain the increase in the electric conductivity of semiconductors with increasing temperature as a result of faster movement of charge carriers or a higher concentration of holes exclusively. Incorrect ideas related to the thermal conductivity of solids are reflected in the idea that heat transfer in a metal occurs through the transfer of vibrational energy of ions of the solid lattice or through the transfer of kinetic energy between free electrons. Furthermore, many students believe that heat transfer in insulators occurs in the same way as in metals, i.e., by free charge carriers.

It seems that most of the above-mentioned erroneous ideas result from the application of non-adequate p-prims or from overly mechanistic reasoning and transferring features of the macroscopic world to the microscopic context. However, many of the identified students' difficulties are a result of suboptimal didactical presentations of solid-state physics. Therefore, the findings from this study may be a good starting point for improving the effectiveness of learning and teaching about microscopic models of thermal and electric conductivity.

In the next step, our aim will be to create and evaluate university-level tutorials on microscopic models of thermal and electric conductivity.

**Author Contributions:** Conceptualization, N.E. and I.A.; methodology, L.J., N.E. and V.M.; validation, L.J., N.E., V.M. and I.A.; formal analysis, L.J., N.E. and I.A.; investigation, N.E. and L.J.; resources, N.E. and L.J.; data curation, N.E. and L.J.; writing—original draft preparation, L.J. and N.E.; writing—review and editing, N.E., L.J., I.A. and V.M.; visualization, L.J. and N.E.; supervision, N.E., I.A. and V.M.; funding acquisition, N.E. and I.A. All authors have read and agreed to the published version of the manuscript.

**Funding:** This research received no external funding.

**Institutional Review Board Statement:** The study was approved by the Ethics Committee of the Faculty of Health Studies, University of Rijeka, Croatia (KLASA:602-04/23-01/105, URBROJ:2170-1-65-23-1, 8 September 2023).

**Informed Consent Statement:** All respondents were informed about the nature of our study and they voluntarily participated in this study. The participants were assured that the principles of confidentiality and anonymity would be adhered to in this study.

**Data Availability Statement:** The data presented in this study are available on request from the corresponding author. The data are not publicly available due to ethical restrictions.

**Acknowledgments:** The authors would like to thank all the participants who took part in this study. This work has been supported in part by the University of Rijeka under the project numbers uniri-pr-prirod-19-5 and uniri-iskusni-prirod-23-33, and also (IA) by the bilateral CRO-SAD project 2/2019.

**Conflicts of Interest:** The authors declare no conflicts of interest.

**Appendix A. The Close-Ended Version of the METCS Concept Inventory**

1. The structure of insulators, conductors and semiconductors as solids can be

    (A) Crystalline or amorphous.
    (B) Crystalline only.
    (C) Crystalline for conductors and semiconductors, and amorphous for insulators.
    (D) Crystalline for insulators and amorphous for conductors and semiconductors.

2. The lattice of the solid insulator consists of

    (A) Atoms connected by covalent bonds.
    (B) Atoms connected by covalent or metallic bonds.
    (C) Atoms connected by covalent or ionic bonds.
    (D) Ions connected by Van der Waals forces.

3. Charge carriers in the insulator

    (A) Are electrons that do not have a pair in a covalent bond.
    (B) Are free electrons.
    (C) Are ions.
    (D) Do not exist.

4. The solid conductor–metal lattice consists of

    (A) Positive ions connected by free electrons.
    (B) Atoms connected by covalent bonds.
    (C) Molecules attracted by Van der Waals forces.
    (D) Positive and negative ions connected by a metal bond.

5. Electrical current carriers in metals are

    (A) Free electrons that form a metallic bond.
    (B) Free excitons.
    (C) Free electrons generated by energetic excitation.
    (D) Free electrons generated under the influence of an electric field.

6. A solid semiconductor lattice consists of

    (A) Electrons and holes.
    (B) Atoms connected by covalent bonds.
    (C) Molecules connected by Van der Waals forces.
    (D) Atoms connected by a metallic bond.

7. Why do free electrons in a pure semiconductor, as opposed to those in a metal, form pairs with holes?

    (A) In pure semiconductors, the mobility of free electrons is lower than in metals.
    (B) Free electrons also form pairs with holes in metals, but they are rarely mentioned.
    (C) The covalent bond between the atoms of a semiconductor is weaker than that of a conductor, which is why holes form in it.
    (D) In metals, there are already free electrons, and in pure semiconductors, they are only created by the excitation of the electron–hole pair.

8. Semiconductors of type N or P are produced by doping pure semiconductors. What is doping?

   (A) The replacement of atoms of the crystal lattice by atoms with a similar electronic structure.
   (B) The addition of holes or protons to a pure semiconductor.
   (C) The addition of conductive substances to a pure semiconductor.
   (D) The addition of electrons to fill holes as a kind of defect in the crystal lattice.

9. Donor impurities that dope the semiconductor are

   (A) Negative electrons.
   (B) Positive ions with a lack of electrons.
   (C) Negative ions with an excess of electrons.
   (D) Neutral atoms with more valence electrons compared to the semiconductor atoms.

10. Acceptor impurities with which the semiconductor is doped are

    (A) Positive ions with a lack of electrons.
    (B) Neutral atoms with fewer valence electrons compared to the semiconductor atoms.
    (C) Positive holes.
    (D) Negative ions with an excess of electrons.

11. What are holes?

    (A) Holes are positively charged real particles with which the semiconductor is doped.
    (B) Holes are empty spaces on some atoms of the crystal lattice that are created by the release of electrons.
    (C) Holes are empty states in the electron band that are created by electron excitation or doping.
    (D) Holes are empty spaces created by electrons on acceptor impurities.

12. What is the graphical representation of the dependence of the mutual potential energy of two atoms of a solid, on their distance?

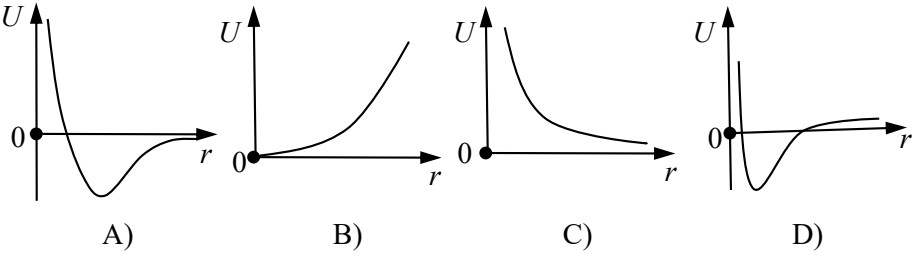

A)                    B)                    C)                    D)

13. Do free electrons collide with each other and/or with the ions of the crystal lattice?

    (A) Free electrons move freely without colliding with each other.
    (B) Free electrons mainly collide with each other like gas molecules.
    (C) Free electrons collide with ions of the crystal lattice that are not in the equilibrium position.
    (D) Free electrons collide with ions in the crystal lattice that are in the equilibrium position.

14. At low temperatures, the collision of free electrons with lattice ions

    (A) Significantly reduced due to the smaller amplitude of ion oscillation.
    (B) Almost unchanged, as the ions are still in their positions in the lattice.
    (C) Reduced due to the lower kinetic energy of the free electrons.
    (D) Unchanged as the free electrons mainly collide with each other.

15. How do free electrons move in a metal?

    (A) They are at rest because no force acts on them.

(B)    They move along atomic orbitals, but at a greater distance from the atomic nucleus.

(C)    They move at high speeds of the order of $10^6$ m/s through the entire crystal.

(D)    They move randomly at low speeds of the order of 1 m/s.

16.    How do electrons move in the conduction band of a semiconductor?

(A)    They are at rest because no force acts on them.

(B)    They move freely like the conduction electrons in a metal.

(C)    They move with restrictions, as the electronic states are more occupied than in a metal.

(D)    They do not move freely, but in a straight line from atom to atom.

17.    How do the holes in a semiconductor move?

(A)    They do not move freely, but in a straight line from atom to atom.

(B)    They are at rest because no external electric field acts on them.

(C)    They move from orbital to orbital without moving towards other atoms.

(D)    They move at high speed like free electrons.

18.    The mobility of the free electrons of a semiconductor in relation to the holes is

(A)    Greater because the holes do not move without the influence of an external electric field.

(B)    The same, because they are the same charge carriers in the semiconductor.

(C)    Higher, because the energy of the electrons in the conduction band is higher than that of the electrons in the valence band.

(D)    Larger in N-type semiconductors and smaller in P-type semiconductors.

19.    The free electrons in a metal move under the influence of an external electric field

(A)    Like a Brownian motion.

(B)    In all directions, whereby the direction opposite to the direction of the electric field predominates.

(C)    Straight in the direction of the electric field.

(D)    In a straight line in the direction opposite to the direction of the electric field.

20.    The light bulb lights up almost instantly after the circuit is closed because the free electrons are set in motion

(A)    Together by an electrical signal travelling at the Fermi speed.

(B)    With a drift velocity that directs their motion and is comparable to the speed of light.

(C)    By the domino effect, in which the electrons collide and produce a common directional motion.

(D)    Almost simultaneously under the effect of an electric field generated at the speed of light.

21.    The following applies to free charge carriers in a semiconductor under the influence of an external electric field

(A)    Electrons move against the direction of the electric field and holes are at rest.

(B)    Electrons and holes move in a straight line in the same direction.

(C)    The electrons move directionally and the holes move randomly.

(D)    Charge carriers move in all directions, with the predominant direction being determined by the electric field.

22.    Adding an acceptor increases the electrical conductivity of the semiconductor by

(A)    Increasing the number of holes.

(B)    Increasing the speed of movement of the holes.

(C)    Increasing the mobility of the electrons in the conduction band.

(D)    Decreasing the energy gap between the valence band and the conduction band.

23. As the temperature rises, the electrical resistance of metals
    (A) Increases due to the stronger vibration of the ions, which hinders the movement of free electrons.
    (B) Decreases because the free electrons move faster.
    (C) Increases because the expansion of the metal means that the free electrons collide less and it is more difficult to transfer current.
    (D) Does not change because the number of free electrons does not change.

24. With increasing temperature, the electrical resistance of a pure semiconductor
    (A) Decreases due to the faster movement of the charge carriers.
    (B) Increases due to a stronger vibration of the ion, which hinders the motion of the charge carriers.
    (C) Changes inversely in relation to metals, due to a higher hole concentration.
    (D) Decreases as the number of free charge carriers increases.

25. Heat transfer in metal occurs mainly
    (A) By the transfer of the vibrational energy of the ions of the solid lattice.
    (B) By the transfer of kinetic energy between free electrons.
    (C) By the transfer of vibrational energy from free electrons to ions.
    (D) By the transfer of kinetic energy from free electrons to ions.

26. Insulators are weaker heat conductors than metals because the heat in them is mainly conducted by
    (A) The vibrations of the lattice atoms, which are weaker due to the stronger interatomic bonds.
    (B) The vibrations of the lattice atoms, but not by free electrons.
    (C) The vibrations of electrons that are not free but are bound to lattice ions.
    (D) Free charge carriers, the number of which is small.

    We bring two solid bodies with different temperatures into contact in a thermally insulated system (see image), whereupon heat is transferred from body A to body B:

$$T_A > T_B$$

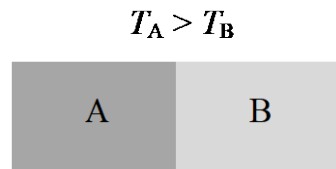

27. At the contact surface, by collision of the molecules of body A and body B
    (A) Mainly intermolecular potential energy is transferred.
    (B) Internal energy is transferred.
    (C) Only the kinetic energy of the molecules is transferred.
    (D) Additional heat is generated.

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
