# Peer review of "Students’ Understanding of Microscopic Models of Electrical and Thermal Conductivity: Findings within the Development of a Multiple-Choice Concept Inventory"

_education, doi:10.3390/educsci14030275_

Round 1

Reviewer 1 Report

Comments and Suggestions for Authors

This manuscript presents meticulously planned and executed research. The authors have implemented excellent procedures for developing and evaluating their concept inventory. The inventory comprehensively addresses topics concerning conductors, insulators, and semiconductors.

The analysis of students' choices of distractors and the explanations behind their selections should prove beneficial for educators teaching these subjects. Additionally, the authors effectively link their findings to existing literature in the field.

Author Response

We thank Reviewer 1 for his/her positive opinion of our research and our manuscript.

Reviewer 2 Report

Comments and Suggestions for Authors

This manuscript describes research that was carefully planned and executed. The authors have followed very good procedures in constructing and testing their concept inventory. The inventory itself seems to cover the topics related to conductors, insulators and semiconductors quite thoroughly.

The discussions of which distractors that the students and comments about why they chose them should be useful for others who are teaching these topics. The authors also do a good job of connecting their results to previous literature in the field.

With a few minor corrections this manuscript is acceptable for publication as it is. The minor issues that I noticed are:

I am not familiar with the term “auditory exercises”. I suspect that it means what Americans call recitations. However, the term should be clarified.

There is a small inconsistency in the punctuation for decimal point. Most of the manuscript use the period (.) but Table 4 uses a comma (,).

A more frequent inconsistency is the use of opening quotation marks. Sometimes the standard English punctuation (“) is used but other times the European („) is used. This issue occurs several times in the manuscript and is particularly noticeable in the following sentences from lines 477-478. something that was taught in class“ [3]. However, this only indicates that they did not “stored it with understanding“, in the first place. (Note that in the latter sentence “stored” should be “store”.

Comments on the Quality of English Language

Minor word usage issues occur in lines 396 and 611 as well as 478 as mentioned in the review.

Author Response

We thank Reviewer 2 for his/her positive opinion about our research and our manuscript. We have taken all his/her suggestions into account; (1) The term "auditory exercises" is clarified in footnote 1 as follows: "Auditory exercises or traditional recitations are tied to the lectures and are led by a teaching assistant who teaches 30 to 50 students. The assistant chooses numerical problems to which the theory from the lectures is applied. Students present their own ideas for possible solutions to the assigned problems, which are modeled on the blackboard." (2) For reasons of consistency in the punctuation of the decimal point, the commas in Table 4 and Figure 1 have been replaced by periods. (3) All quotation marks have been replaced with punctuation (") and ("). (4) The word "stored" has been replaced by the word "store".